# Antimicrobial Resistance in Enterobacterales Recovered from Urinary Tract Infections in France

**DOI:** 10.3390/pathogens11030356

**Published:** 2022-03-15

**Authors:** Eric Farfour, Laurent Dortet, Thomas Guillard, Nicolas Chatelain, Agathe Poisson, Assaf Mizrahi, Damien Fournier, Rémy A. Bonnin, Nicolas Degand, Philippe Morand, Frédéric Janvier, Vincent Fihman, Stéphane Corvec, Lauranne Broutin, Cécile Le Brun, Nicolas Yin, Geneviève Héry-Arnaud, Antoine Grillon, Emmanuelle Bille, Hélène Jean-Pierre, Marlène Amara, Francoise Jaureguy, Christophe Isnard, Vincent Cattoir, Tristan Diedrich, Emilie Flevin, Audrey Merens, Hervé Jacquier, Marc Vasse

**Affiliations:** 1Service de Biologie Clinique, Hôpital Foch, 92150 Suresnes, France; m.vasse@hopital-foch.com; 2Team RESIST, Laboratoire de Bactériologie-Hygiène, Assistance Publique des Hôpitaux de Paris, Faculté de Médecine, CHU de Bicêtre, Université Paris-Saclay, UMR 1184, 95270 Le Kremlin-Bicêtre, France; laurent.dortet@aphp.fr (L.D.); remy.bonnin@u-psud.fr (R.A.B.); 3Inserm UMR-S 1250 P3Cell, SFR CAP-Santé, Laboratoire de Bactériologie-Virologie-Hygiène Hospitalière-Parasitologie-Mycologie, Hôpital Robert Debré, CHU Reims, Université de Reims-Champagne-Ardenne, 51000 Reims, France; tguillard@chu-reims.fr; 4BIOPATH Laboratoires, 62231 Coquelles, France; chatelain.bio@free.fr; 5Laboratoire BIO ARD’AISNE, 08362 Rethel, France; agathe.poisson@bioardaisne.fr; 6Service de Microbiologie Clinique, Groupe Hospitalier Paris Saint-Joseph, 75015 Paris, France; amizrahi@ghpsj.fr; 7Institut Micalis UMR 1319, Université Paris-Saclay, Institut National de Recherche pour l’Agriculture, l’Alimentation et l’Environnement, AgroParisTech, 92290 Châtenay Malabry, France; 8Centre National de Référence de la Résistance aux Antibiotiques, Centre Hospitalier Universitaire de Besançon, 25000 Besançon, France; dfournier@chu-besancon.fr; 9Laboratoire de Bactériologie, Centre Hospitalier Universitaire de Nice, 06200 Nice, France; nicolas.degand@ch-antibes.fr; 10Service de Bactériologie, AP-HP Centre-Université de Paris, Site Cochin, 75014 Paris, France; philippe.morand@aphp.fr; 11Service de Biologie, HIA Sainte-Anne, 83000 Toulon, France; janvierfrede@gmail.com; 12Bacteriology and Infection Control Unit, Department of Prevention, Diagnosis, and Treatment of Infections, AP-HP Centre, Henri-Mondor University Hospital, 94000 Creteil, France; vincent.fihman@aphp.fr; 13Inserm, Service de Bactériologie et des Contrôles Microbiolgoiques, CHU de Nantes, Université de Nantes, 44000 Nantes, France; stephane.corvec@chu-nantes.fr; 14Service de Bactériologie et d’Hygiène Hospitalière, Unité de Microbiologie Moléculaire et Séquençage, CHU de Poitiers, 86000 Poitiers, France; lauranne.broutin@chu-poitiers.fr; 15Service de Bactériologie, Virologie et Hygiène Hospitalière, CHU de Tours, 37000 Tours, France; c.lebrun@chu-tours.fr; 16Department of Microbiology, Laboratoire Hospitalier Universitaire de Bruxelles—Universitair Laboratorium Brussel (LHUB-ULB), Université Libre de Bruxelles (ULB), 1000 Brussels, Belgium; nicolas.yin@lhub-ulb.be; 17Department of Microbiology, Institut Gustave Roussy, Université Paris-Saclay, 94800 Villejuif, France; 18Inserm UMR 1078 GGB, Unité de Bactériologie, Hôpital La Cavale Blanche, CHRU de Brest, Université de Brest, CEDEX, 29609 Brest, France; genevieve.hery-arnaud@univ-brest.fr; 19Fédération de Médecine Translationnelle de Strasbourg, Institut de Bactériologie, Université de Strasbourg, VBP EA7290, 67000 Strasbourg, France; antoine.grillon@hotmail.fr; 20Service de Microbiologie, Assistance Publique-Hôpitaux de Paris, Hôpital Necker Enfants-Malades, AP-HP Centre-Université de Paris, 75015 Paris, France; emmanuelle.bille@aphp.fr; 21Laboratoire de Bactériologie, Centre Hospitalier Universitaire de Montpellier, 34000 Montpellier, France; h-jean_pierre@chu-montpellier.fr; 22Maladies Infectieuses et Vecteurs—Écologie, Génétique, Évolution et Contrôle, Centre National pour la Recherche Scientifique, Institut de Recherche pour le Développement, Université de Montpellier, 34000 Montpellier, France; 23Service de Biologie, Unité de Microbiologie, CH de Versailles, 78150 Le Chesnay, France; mamara@ch-versailles.fr; 24Service de Microbiologie Clinique, Groupe Hospitalier Paris Seine Saint-Denis, AP-HP Centre, CHU Avicenne, 93000 Bobigny, France; francoise.jaureguy@aphp.fr; 25Department of Microbiology, CHU de Caen Normandie, Normandie University, UNICAEN, 14000 Caen, France; isnardc@me.com; 26Service de Bactériologie-Hygiène, CHU de Rennes, 35033 Rennes, France; vincent.cattoir@chu-rennes.fr; 27Service de Microbiologie, CH de Valenciennes, 59300 Valenciennes, France; diedrich-t@ch-valenciennes.fr; 28Laboratoire de Biologie, CH de Dieppe, 76200 Dieppe, France; eflevin@ch-dieppe.fr; 29SSA (French Military Health Service), Bégin Military Teaching Hospital, 94160 Saint-Mandé, France; merens-a@wanadoo.fr; 30Service de Bactériologie-Virologie, AP-HP Centre, Hôpital Lariboisière, 75010 Paris, France; herve.jacquier@aphp.fr

**Keywords:** Enterobacterales, ESBL, urinary tract infection (UTI), fosfomycin, carbapenem, temocillin, nitrofurantoin, pivmecillinam

## Abstract

In the context of increasing antimicrobial resistance in Enterobacterales, the management of these UTIs has become challenging. We retrospectively assess the prevalence of antimicrobial resistance in Enterobacterales isolates recovered from urinary tract samples in France, between 1 September 2017, to 31 August 2018. Twenty-six French clinical laboratories provided the susceptibility of 134,162 Enterobacterales isolates to 17 antimicrobials. The most frequent species were *E. coli* (72.0%), *Klebsiella pneumoniae* (9.7%), *Proteus mirabilis* (5.8%), and *Enterobacter cloacae* complex (2.9%). The overall rate of ESBL-producing Enterobacterales was 6.7%, and ranged from 1.0% in *P. mirabilis* to 19.5% in *K. pneumoniae*, and from 3.1% in outpatients to 13.6% in long-term care facilities. Overall, 4.1%, 9.3% and 10.5% of the isolates were resistant to cefoxitin, temocillin and pivmecillinam. Cotrimoxazole was the less active compound with 23.4% resistance. Conversely, 4.4%, 12.9%, and 14.3% of the strains were resistant to fosfomycin, nitrofurantoin, and ciprofloxacin. However, less than 1% of *E. coli* was resistant to fosfomycin and nitrofurantoin. We identified several trends in antibiotics resistances among Enterobacterales isolates recovered from the urinary tract samples in France. Carbapenem-sparing drugs, such as temocillin, mecillinam, fosfomycin, cefoxitin, and nitrofurantoin, remained highly active, including towards ESBL-E.

## 1. Introduction

Urinary tract infections (UTIs), including community-acquired and healthcare-associated infections, are the most frequent infections caused by Enterobacterales [1]. In the context of increasing antimicrobial resistance in Enterobacterales, the management of these UTIs has become challenging. The worldwide spread of extend-spectrum β-lactamase (ESBL)-producing Enterobacterales (ESBL-E), especially in *Escherichia coli*, is of particular concern because of its spread in the community. In France, 3.3% of community-acquired urinary tract infections (CA-UTIs) are due to ESBL-E [2], but ESBL-E could reach 40% of Enterobacterales isolates in some countries [3,4]. In addition, since 2010s, the spread of carbapenemase-producing Enterobacterales (CPE) has over-challenged the management of infections caused by Enterobacterales in areas of high CPE prevalence.

Empirical treatment is recommended for UTIs according to the antibiotic resistance risk level adapted for the clinical criteria. Accordingly, ≤20% of resistant isolates are required to accept an empirical treatment for uncomplicated cystitis but a prevalence of resistant isolates have to be ≤10% for all other CA-UTIs [5,6,7]. Since the prevalence of resistance might change over time depending on several factors such as the use of antibiotics or any changes in the bacterial epidemiology, the guidelines for the management of CA-UTIs require regular updates.

The aim of the study was to assess the prevalence of antimicrobial resistance in Enterobacterales isolates recovered from urinary tract samples in France. The GMC-12 study was carried out by the GMC (Groupe de Microbiologie Clinique) study group, a collaborative association of 40 French clinical microbiologists involved in clinical research around the country.

## 2. Results

### 2.1. Bacterial Species

Overall, 134,162 clinical isolates were included from 1 September 2017, to 31 August 2018. The median number of isolates included by each center was 4322 (interquartile range 2587–5421). Enterobacterales species recovered are listed in Appendix A, briefly, 72.0% *E. coli*, 9.7% *Klebsiella pneumoniae* complex, 5.8% *Proteus mirabilis*, 2.9% *Enterobacter cloacae* complex, 2.3% *Citrobacter koseri*, 1.9% *Klebsiella oxytoca*, 1.5% *Morganella morganii*, and 3.9% other Enterobacterales species. The species diversity was variable depending on the ward of the sample collection (Appendix A).

*E. coli* accounted for more than 75% of all enterobacterial isolates in five wards: psychiatry (85.5%), obstetrics/gynecology (80.9%), pediatrics (78.2%), outpatient (76.4%), and emergency (76.1%). The prevalence of *E. coli* was lower in long term care facilities (LTCFs), surgery and intensive care units (ICUs) with rates of 63.2%, 61.3%, and 63.1%, respectively. As expected, the diversity of Enterobacterales species was lower in wards with the highest prevalence of *E. coli* (Appendix A). For all wards, a single or two species accounted for at least 10% of all positive urines (i.e., in all cases *E. coli* and sometimes *K. pneumoniae*). In obstetrics/gynecology and psychiatry, less than 5 species represented at least 1.0% of all positive urine samples, while at least 9 species accounted for at least 1.0% of the total in surgical departments, LTCFs, medicine, oncology/hematology departments, and ICUs.

### 2.2. ESBL-Producing Enterobacterales (ESBL-E)

The overall rate of ESBL-E was 6.7%, with significant differences according to the bacterial species, the geographic regions, or the wards. The prevalence of ESBL-E ranges from 1.0% in *P. mirabilis* to 19.5% in *K. pneumoniae* complex (Figure 1). It was more than 10% in two species, *K. pneumoniae* complex (19.5%) and *E. cloacae* (18.2%); between 5% and 10% in two other species, *E. coli* (5.5%) and *C. freundii* (5.9%); and below 5% in all others species, including *P. mirabilis* (1.0%), *C. koseri* (1.8%), and *K. oxytoca* (3.7%). Depending on the geographic region, the rate of ESBL-E ranged from 3.8% in Centre Val de Loire to 9.7% in Ile-de-France (Figure 2). ESBL-E were more frequently recovered from patients sampled in LTCFs (13.6%) and oncology/hematology wards (9.5%), whereas less than 5.0% of the isolates were ESBL-E in three departments (outpatients (3.1%), obstetrics/gynecology (3.5%), and pediatrics (4.5%)) (Figure 3).

### 2.3. Resistance to β-Lactams

Acquired resistance to amoxicillin was observed in 50.9% and 39.2% of *E. coli* and *P. mirabilis*, respectively (Figure 4). Overall, 26.4% of the Enterobacterales isolates intrinsically susceptible to amoxicillin/clavulanate were resistant to this combination. Resistance to amoxicillin/clavulanate reached 28.6% both in *E. coli* and *K. pneumoniae* while it remained below 10% in *P. vulgaris* (8.4%) and *C. koseri* (3.5%). Overall, 12.1% of the isolates were resistant to piperacillin/tazobactam. The highest prevalences of resistance were found in *E. cloacae* complex (41.3%), *C. freundii* (31.4%), *K. aerogenes* (27.2%) and *K. pneumoniae* (22.8%). In contrast, only 9.9% of *E. coli*, 2.8% of *C. koseri* and 1.6% of *P. mirabilis* were resistant to piperacillin/tazobactam.

The overall rate of resistance to temocillin and pivmecillinam was about 10% (Figure 5). In *E. coli*, resistance to temocillin and pivmecillinam was found in 9.5% and 8.6% of the isolates, respectively. Pivmecillinam displayed lower activity against *Proteus* spp., *S. marcescens* and *M. morganii* with resistance rates of >25.0%, 69.2%, and 70.5%, respectively. In addition, cephalosporinase-overproducing species were more likely resistant to temocillin. For instance, 37.8% of *S. marcescens*, 26.8% of *E. cloacae* complex, 19.5% of *C. freundii*, and 12.5% of *K. aerogenes* were not susceptible to temocillin.

Cefoxitin was highly active towards all intrinsically susceptible species (overall resistance rate at 4.1%). Regarding 3rd-generation cephalosporins (3GC), decreased susceptibility was observed in 9.3% of the isolates. This rate of 3GC decreased susceptibility was higher in cephalosporinase-overproducing species and reached up to 46.6%, 35.2%, and 27.5% in *E. cloacae* complex, *C. freundii*, and *K. aerogenes*, respectively. The overall resistance rates to cefepime and aztreonam were high, 16.0% and 20.6%, respectively. However, both antibiotics were tested only for a part of isolates included: 44,905 isolates (33.5%) for cefepime and 34,572 isolates (25.8%) for aztreonam, mainly those resistant to 3GC. Resistance to imipenem was very low (<0.5%) in all species, except *C. freundii* (1.1%), *K. aerogenes* (1.0%), and *E. cloacae* complex (0.9%). However, 13.8%, 3.4%, and 2.8% of *E. cloacae* complex, *K. aerogenes*, and *C. freundii* isolates were found to be resistant to ertapenem, respectively.

Of note, *C. koseri* and *Proteus* spp. were the enterobacterial species with the highest susceptibility to all β-lactams, with less than 5% of them displaying a decreased susceptibility to any β-lactam. An exception has to be noted for mecillinam and amoxicillin-clavulanate in *P. mirabilis* (rate of resistance of 26.8% and 10.5%, respectively) and *P. vulgaris* (rate of resistance of 31.8% and 8.4%, respectively).

### 2.4. Resistance to Other Classes of Antibiotics

Among non-β-lactam antibiotics, cotrimoxazole was the least active compound with ~25% resistance in isolates of *E. coli*, *K. pneumoniae*, *E. cloacae* complex, and *P. mirabilis* (Figure 5a). Only *C. koseri* and *K. aerogenes* displayed <5.0% resistance rates to this molecule. Intermediate resistance rates from 5.0% to 10.0% were observed for *K. oxytoca* and *S. marcescens* while this cotrimoxazole-resistance prevalence reached at least 10.0% in all other species.

Amikacin was highly active against all Enterobacterales species, with an overall resistance rate of only 2.5%, reaching a maximum of 4.9% in *E. cloacae* complex (Figure 5b). The rate of resistance to gentamicin was higher, with four species displaying a resistance prevalence over 10%: *E. cloacae* complex (24.0%), *C. freundii* (14.0%), *K. pneumoniae* complex (13.8%), and *P. mirabilis* (11.5%). The rate of *E. coli* isolates resistant to nitrofurantoin or fosfomycin was low (~1.0% for each drug), contrary to *K. pneumoniae* and *E. cloacae* complex, which displayed resistance rates greater than 25% and 14% for nitrofurantoin and fosfomycin, respectively (Figure 5a).

Regarding quinolones, the resistance rate was lower for ciprofloxacin (14.3%) than for ofloxacin (16.7%) and nalidixic acid (18.5%) (Figure 5c). Resistance to all three molecules was below 5% in *C. koseri* and *P. vulgaris*, between 5% and 10% in *K. oxytoca* and *S. marcescens*, and over 10% in all other species, including *E. coli*.

### 2.5. Associated Resistance

The prevalence of associated resistance was calculated for the 7 leading species and 8 antibiotics (Table 1). Associated resistances were systematically below 0.5% in *P. mirabilis* and C. koseri, regardless of the drugs. They were slightly higher in *M. morganii*, *K. oxytoca*, and *E. coli*, but almost always below 5%, except for aztreonam and gentamicin in *E. coli* (5.7%). Associated resistances were much higher in *K. pneumoniae* and *E. cloacae* complex. Overall, 26.1% of *E. cloacae* complex and 16.6% of *K. pneumoniae* isolates were resistant to both 3GC, cotrimoxazole, and ciprofloxacin. Less than 5% of *K. pneumoniae* and *E. cloacae* complex isolates were resistant to both 3GC and amikacin, while 12.8% and 23.4% were resistant to both 3GC and gentamicin respectively. A similar prevalence of resistance was noticed for aztreonam and amikacin or gentamicin.

## 3. Discussion

Our results highlight several heterogeneities in both species distribution and rates of antimicrobial resistances among Enterobacterales isolates collected from urinary tract samples in France. ESBL-E were more frequent among *K. pneumoniae* and *E. cloacae* and also, in oncologic/hematologic and intensive care wards. Furthermore, there was a strong disparity among French regions. Carbapenem-sparring alternatives, such as cefoxitin, fosfomycin, nitrofurantoin, temocillin, and pivmecillinam, remained highly active against most Enterobacterales species.

As expected, *E. coli* was the leading species recovered from UTIs regardless the context. Regarding outpatients, *E. coli* accounted for about 80% of all Enterobacterales isolates collected from urinary tract samples, which was similar to other countries [8,9,10]. We found a similar distribution of Enterobacterales species in obstetrics/gynecology, pediatrics, and psychiatry as in outpatients. We also described a lower diversity of bacterial species recovered from these patients reflecting a similar bacterial ecology in comparison to the community. Indeed, all these three wards are usually characterized by either short hospital stays, or few comorbidities, or infrequent history of antibiotic treatment during the past months. All these factors were previously associated with an increased risk of antimicrobial resistance [11,12]. These findings suggested that the empiric management of UTIs could be similar for outpatients and those of obstetrics/gynecology, pediatrics, and psychiatry without any known risk factor for antimicrobial resistance. In contrast, a higher diversity of species was recovered in urinary samples of patients admitted in all other departments, including LTCFs. In these departments, a higher number of Enterobacterales species usually considered to be more frequently involved in hospital-acquired infections and more frequently responsible for complicated UTIs, such as *Enterobacter* spp. or *Klebsiella* spp., are isolated [1]. The prevalence of Enterobacterales species was similar in LTCFs, to medicine and surgery wards. These similarities could be explained by extrinsic findings, LTCFs are downstream institutions for medicine and surgery wards, but also by intrinsic findings, such as a common dining room.

In addition, the rate of ESBL-E was also higher in these typically two nosocomial species (*Enterobacter* spp. or *Klebsiella* spp.). Both species are widely distributed in the natural environment and the gastrointestinal tract of a wide range of animals, which might explain their greater ability to survive in the hospital environment and to cause outbreaks [13]. The rates of ESBL-E among Enterobacterales were heterogeneously distributed throughout French regions. The two regions with the higher rates of ESBL-E, “Ile-de-France” and “PACA”, were those with the highest densities of inhabitants. In addition, ESBL-E were more frequently isolated from urine samples of LTCF residents, confirming they could be a reservoir for antimicrobial resistance [4]. Furthermore, the prevalence of Enterobacterales species is similar in LTCFs, medicine and surgery wards suggesting a similar epidemiology. Regarding this high rate of ESBL-E and the frequency of urinary tract colonization, high adherence to antibiotic stewardship and infection prevention and control measures is required in LTCFs. At the opposite, the rate of ESBL-E remained low in outpatients (3.1%), as previously reported [2,14]. Resistance to 3GC remains below 10% (9.3%), confirming that these drugs can still be used for complicated UTIs.

Cefoxitin, considered as a carbapenem-sparing alternative for the treatment of UTIs, was one of the most active β-lactams. In France, while evidence of efficiency was shown in male UTIs [15,16], cefoxitin is mainly recommended for the treatment of uncomplicated female UTIs due to ESBL-producing *E. coli* [6]. Interestingly, Enterobacterales displayed moderate resistance rates towards temocillin and mecillinam. Regarding these two molecules, the highest rates of resistance were observed in cephalosporinase-overproducing species, such as *E. cloacae* complex, *S. marcescens*, and *K. aerogenes*. Despite the fact that resistance mechanisms to temocillin remain unknown, it was suggested that the drug might be hydrolyzed by high-level cephalosporinase [17,18]. However, we could not exclude that temocillin resistance may be due to a combination of still-unknown determinants. Imipenem was almost always active against Enterobacterales isolates, reflecting the low prevalence of CPE in France both in the community and in hospital settings [19]. However, the rate of ertapenem non-susceptible isolates was high in *E. cloacae* complex, *K. aerogenes*, and *C. freundii*. In these species, resistance to ertapenem is usually related to cephalosporinase-overproduction associated with decreased permeability of the outermembrane [20,21]. Emergence of cephalosporinase-overproducing isolates has been described to be related to consumption of 3GC [22,23]. It is noteworthy that among European countries, antibiotic consumption is higher in France [24].

Interestingly, *E. coli* remained highly susceptible to fosfomycin and nitrofurantoin suggesting the absence of antimicrobial resistance reservoir for these drugs in both hospital settings and outpatients. The rate of resistance to fosfomycin was higher in *Klebsiella* species than in other Enterobacterales. However, in harmonization with the EUCAST guidelines, the breakpoint of fosfomycin in Enterobacterales was updated in the 2019 version of the CA-SFM (inhibition diameter of 24 mm vs. 19 mm with the disc diffusion method) [25]. Consequently, the rate of fosfomycin resistance might be underestimated in the present study. Indeed, with updated breakpoints, the rate of resistance to fosfomycin was reported to increase of about 3-fold for all Enterobacterales species, reaching 80% in *Klebsiella* spp. [26].

As conclusion, we identified several trends in species distribution and antibiotics resistances among Enterobacterales isolates recovered from the urinary tract samples in France. Carbapenem-sparing drugs, such as temocillin, mecillinam, fosfomycin, cefoxitin, and nitrofurantoin remained highly active, including towards ESBL-E. We highlighted a strong heterogeneity among French regions in the prevalence of ESBL-E, as well as between enterobacterial species and hospitalization wards. This ESBL-E prevalence remained low in outpatients (3.1%) whereas it could reach up to 13.6% in LTCFs. Complementary analysis including clinical and socio-demographic findings (i.e., age and sex, and antibiotic consumption) would help to highlight and to amore, data regarding the sex and the age of the patients were not collected and we were, therefore, not able to identify. Indeed, due to distinct physiopathology and recommended antibiotic treatment according to the gender and age, the trends in antibiotic resistance could be different among the patients [8,27]. In the same way, the characterization of mechanisms of antibiotic resistance might enhance the comprehension of their diffusion.

## 4. Materials and Methods

Twenty-six French clinical laboratories spread over the country participated in this retrospective study. Except for the Auvergne-Rhône Alpes region, at least one center was located in each of the 15 metropolitan French regions (Appendix A).

All Enterobacterales isolates collected from urinary samples between 1 September 2017 and 31 August 2018 were included. A single isolate exhibiting the same antibiotic susceptibility profile was included per patient. For each isolate, the medical ward where the patient was admitted during sample collection and the susceptibility to 17 antimicrobials were recorded. The medical wards were classified as follows: surgery, LTCFs, obstetrics/gynecology, medicine, oncology/hematology, pediatrics, psychiatry, ICU, emergency, and outpatients.

Bacterial identification was routinely performed using conventional biochemical methods (e.g., VITEK 2, bioMérieux) or MALDI-TOF mass spectrometry as recommended by the manufacturers. As *K. pneumoniae* and *K. variicola* could not be distinguished using biochemical methods, they were grouped within the *K. pneumoniae* complex. Antimicrobial susceptibility testing (AST) was performed and interpreted according to the CA-SFM/EUCAST 2017 v1.0 guidelines [28]. A phenotypic-based approach was used to distinguish ESBL-E and cephalosporinase-overproducing Enterobacterales isolates, as previously described [28]. The distribution of bacterial species was analyzed according to the French region and admission wards. The rate of isolates with a decreased susceptibility was calculated for each antibiotic according to bacterial species. Intrinsic resistance to antibiotics were excluded from the analysis.

Statistical analysis were performed using R software (R-Core Team) [29]. Categorical and continuous variables were compared using the Chi-square and the Student’s test, respectively.

## Figures and Tables

**Figure 1 pathogens-11-00356-f001:**
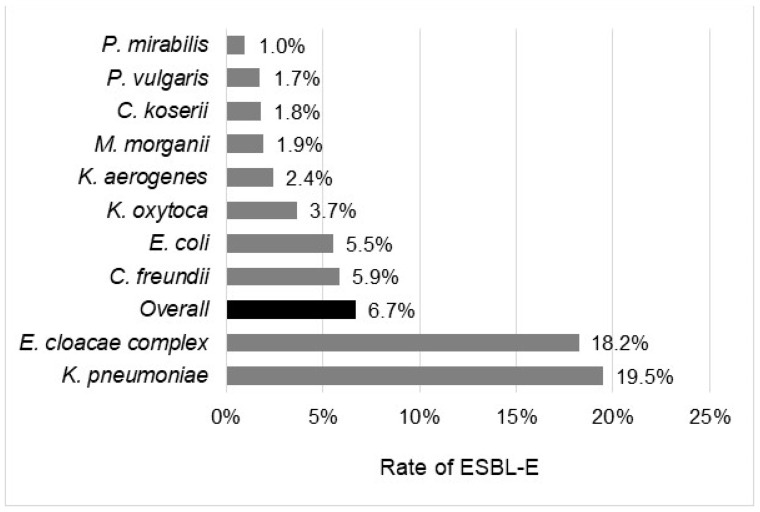
Rate of ESBL-E according to the bacterial species.

**Figure 2 pathogens-11-00356-f002:**
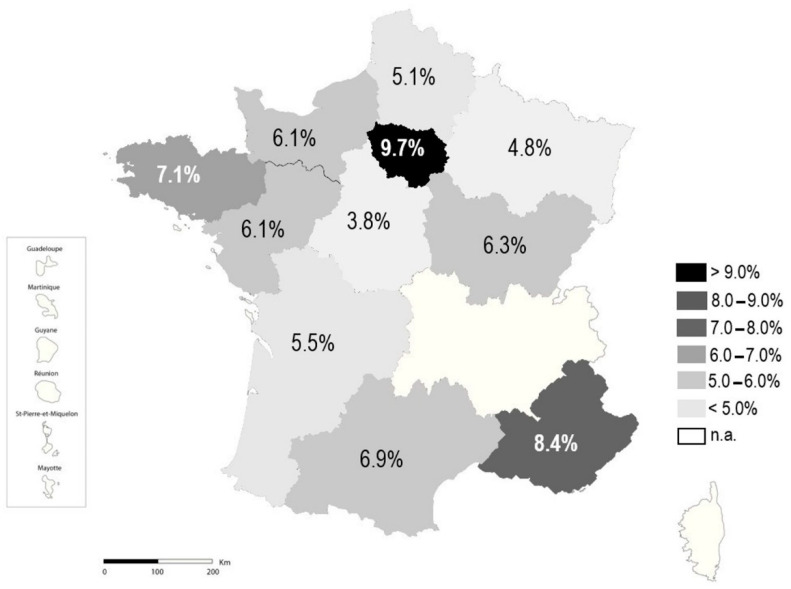
Rate of ESBL-E according to region.

**Figure 3 pathogens-11-00356-f003:**
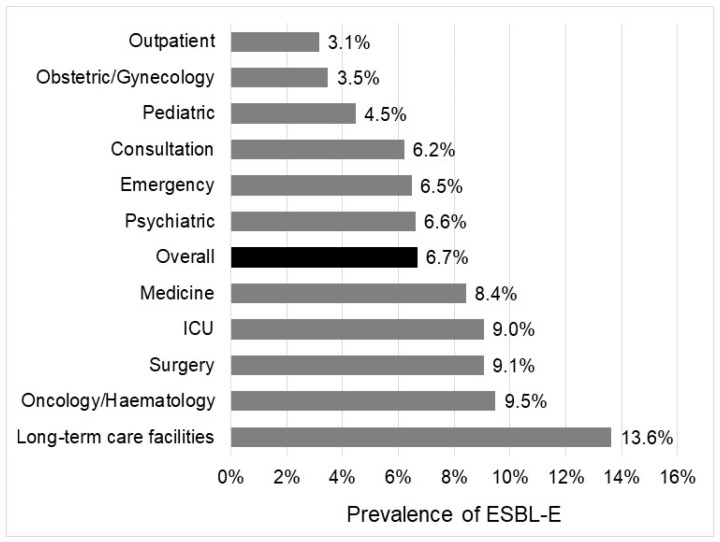
Rate of ESBL-E according to ward of sampling.

**Figure 4 pathogens-11-00356-f004:**
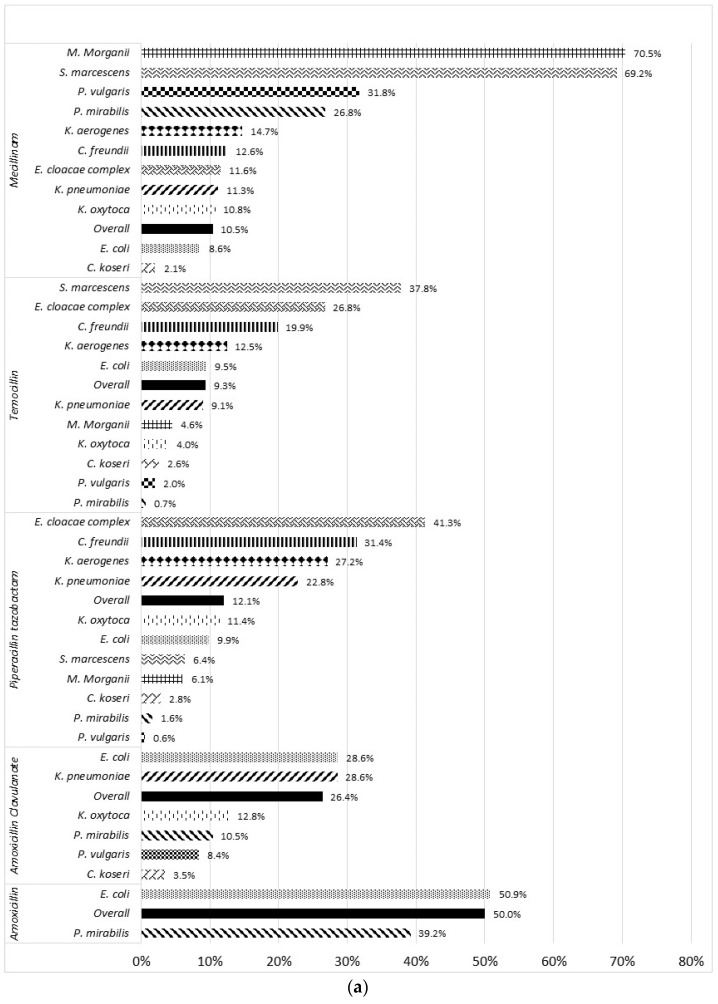
Rate of resistance to 9 β-lactam antibiotics according to bacterial species. (**a**) Penicillin and (**b**) cephalosporin and monobactam.

**Figure 5 pathogens-11-00356-f005:**
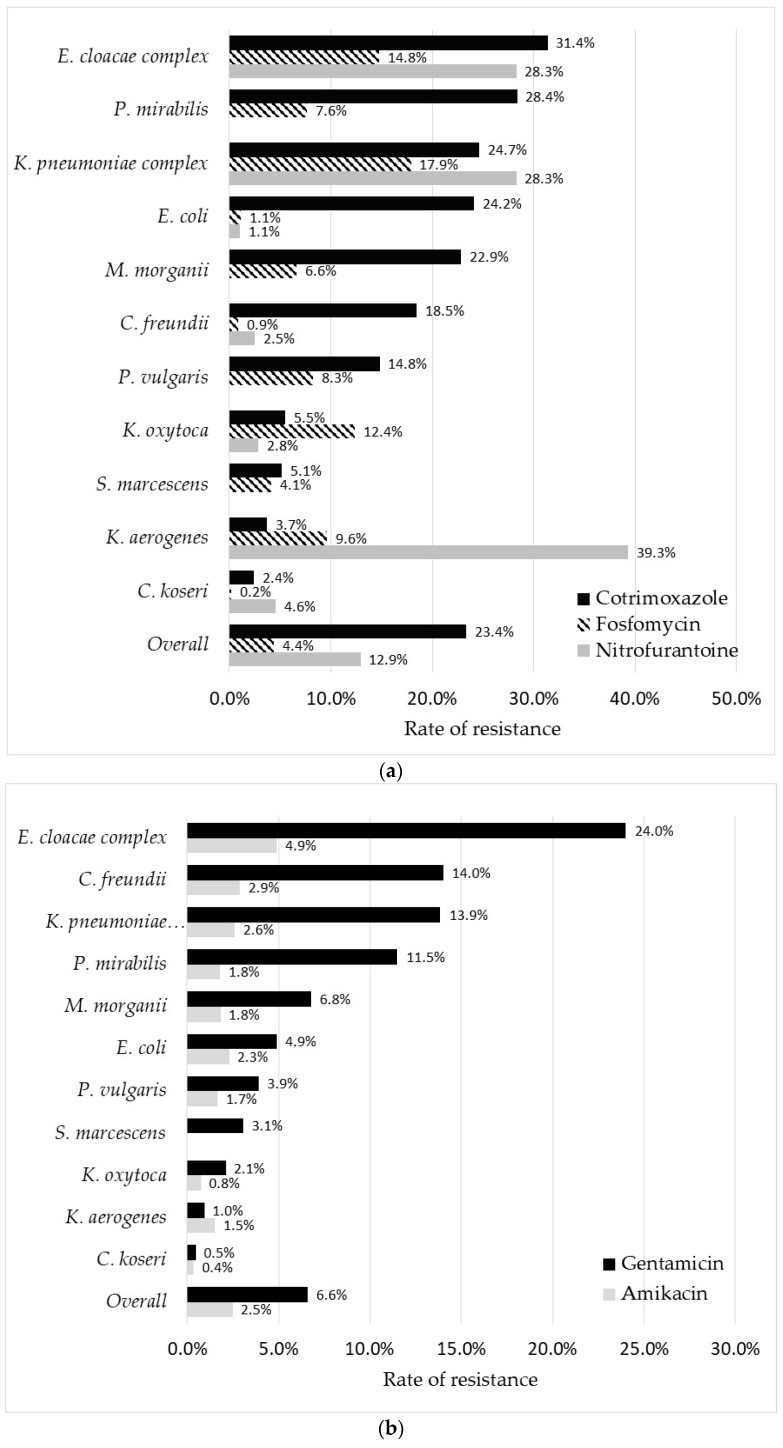
Rate of resistance to 8 non-β-lactam antibiotics according to bacterial species. (**a**) Cotrimoxazole, Fosfomycin, and nitrofurantoin, (**b**) gentamicin and amikacin, and (**c**) nalidixic acid, ofloxacin, and ciprofloxacin.

**Table 1 pathogens-11-00356-t001:** Prevalence of associated resistance in 7 Enterobacterales species. 3GC: 3rd generation cephalosporin; SXT: cotrimoxazole; CIP: Ciprofloxacin; FEP: Cefepime; TEM: temocillin; AMI: Amikacin; GENTA: Gentamicin; AZT: Aztreonam.

	*E. coli*	*K. pneumoniae*	*E. cloacae* complex	*M. morganii*	*K. oxytoca*	*P. mirabilis*	*C. koseri*
3GC + SXT + CIP	3.0%	16.6%	26.1%	3.8%	1.6%	0.5%	0.5%
PTZ + SXT + CIP	2.1%	12.1%	23.9%	0.8%	1.4%	0.2%	0.1%
TEM + SXT + CIP	2.3%	6.9%	15.0%	1.1%	1.3%	0.1%	0.2%
3GC + AMI	0.7%	2.3%	4.7%	0.7%	0.6%	0.0%	0.2%
3GC + GENTA	1.7%	12.8%	23.4%	1.6%	1.8%	0.3%	0.3%
AZT + AMI	2.6%	3.8%	3.4%	0.5%	0.8%	0.0%	0.1%
AZT + GENTA	5.7%	17.4%	26.5%	1.1%	2.3%	0.2%	0.2%

## Data Availability

The data presented in this study are available on request from the corresponding author.

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
