# Peer review of "Antimicrobial Resistance in Enterobacterales Recovered from Urinary Tract Infections in France"

_pathogens, 2022, doi:10.3390/pathogens11030356_

Round 1
Reviewer 1 Report
Line 102: Reword the sentence
Line 111: Why do you think the prevalence of E.coli was lower in long-term facilities? Maybe include this in the discussion.
Discussion: May a brief outline of limitations of the study
Reviewer 2 Report
This paper aims to assess the prevalence of antibiotic resistance across extended spectrum beta lactamase (ESBL) producing bacteria when sampled against a large population in France including different wards of hospital from 2017 till 2018. As expected, Escherichia coli is the most prevalent species accounting for 72 percent of the total cases and the leading cause of UTI across all the wards: psychiatry, obstetrics/gynecology, pediatrics, outpatients, and emergency. However, E. cloacae complex and Klebsiella pneumoniae not E. coli showed greatest resistance against the different antibiotics the authors tested. Interestingly, the greatest diversity of antibiotic resistance was found in long term care facilities suggesting the presence of reservoir of antibiotic resistance whereas the lowest resistance was found in outpatients department which are typically associated with short time hospitalizations and least antibiotic treatment in the recent medical history of the patients. The rate of ESBL-E among E. coli was found be significantly lower (6.7%) as compared to Klebsiella pneumoniae (19.5%). The rate of ESBL-E showed significant difference across difference bacterial species. 50.9% E. coli showed resistance to amoxicillin and 28.6% showed resistance to combinational drugs treatment of Amoxicillin/Clavulanate, 9.7% to Temocillin and 8.6% to Pivmecillinam, which is close to the median (10%) resistance across Enterobacterales. E. cloacae showing significantly increased resistance of 52% to AZT, 46% to 3GC and 38% to Cefepime as compared to 19.3%, 6.4% and 15% resistance in E. coli respectively. The rate of resistance against non-beta-lactamase antibiotics was found to be greatest across E. cloacae complex with 31.4%, 14.8% and 28.3% resistance against Cotrimoxazole, Fosfomycin and Nitrofurantoine although the highest resistance against Nitrofurantoine was found among K. aerogenes (39%). E. cloacae complex also showed highest resistance against Gentmaycin, Amikacin, Ciprofloxacin (31.6%), Ofloxacin (32.3%), Nalidixic Acid (34.4%). Which is significantly greater than the overall resistance across the Enterobacterales. Overall, this paper highlights the trends that are observed across different species of Enterbacterales in reference to their resistance against broadly used antibiotics. The paper is well written and should of interest to broader audience.
However, I have a couple of comments that the authors should address
Minor comments
- The authors should discuss about the resistance percentage of E. cloacae and Klebsiella pneumoniae when subjected to combinational drug therapy available to treat complicated UTI. Is there any strain of bacteria they isolated that should resistance to multiple antibiotics when treated in sync?
- The authors have mentioned that ICU patients showed a resistance of 9% for ESBL-E which is significantly greater than the outpatients (3.1%) and the overall median (6.7%). Are these typically isolated from patients showing acute form of sepsis due to UTI or these are isolated with patients having multiple symptoms of other diseases along with UTI. The authors should comment on that.
- The authors should comment on frequencies of overlap of different bacterial species that were isolated with people diagnosed with UTI. For example, what percentage of people with UTI had both Escherichia coli and Klebsiella pneumoniae or Enterobacter cloacae in their body and so forth. They could use a Venn diagram to illustrate the point.
